# Human Anelloviruses: Influence of Demographic Factors, Recombination, and Worldwide Diversity

María Cebriá-Mendoza,[a] Beatriz Beamud,[a,b] Iván Andreu-Moreno,[a] Cristina Arbona,[c] Luís Larrea,[c] Wladimiro Díaz,[a,d,e] Rafael Sanjuán,[a,f] José M. Cuevas[a,e]

[a]Institute for Integrative Systems Biology (I2SysBio), Universitat de València-CSIC, Valencia, Spain
[b]FISABIO-Salud Pública, Generalitat Valenciana, Valencia, Spain
[c]Centro de Transfusión de la Comunidad Valenciana, Valencia, Spain
[d]Genomic and Health Area, Foundation for the Promotion of Sanitary and Biomedical Research of the Valencia Region (FISABIO), Valencia, Spain
[e]Centro de Investigación Biomédica en Red en Epidemiología y Salud Pública (CIBEResp), Madrid, Spain
[f]Department of Genetics, Universitat de València, Valencia, Spain

**ABSTRACT** Anelloviruses represent the major and most diverse component of the healthy human virome, referred to as the anellome. In this study, we determined the anellome of 50 blood donors, forming two sex- and age-matched groups. Anelloviruses were detected in 86% of the donors. The number of detected anelloviruses increased with age and was approximately twice as high in men as in women. A total of 349 complete or nearly complete genomes were classified as belonging to torque teno virus (TTV), torque teno mini virus (TTMV), and torque teno midi virus (TTMDV) anellovirus genera (197, 88, and 64 sequences, respectively). Most donors had intergenus (69.8%) or intragenus (72.1%) coinfections. Despite the limited number of sequences, intradonor recombination analysis showed 6 intragenus recombination events in ORF1. As thousands of anellovirus sequences have been described recently, we finally analyzed the global diversity of human anelloviruses. Species richness and diversity were close to saturation in each anellovirus genus. Recombination was found to be the main factor promoting diversity, although its effect was significantly lower in TTV than in TTMV and TTMDV. Overall, our results suggest that differences in diversity between genera may be caused by variations in the relative contribution of recombination.

**IMPORTANCE** Anelloviruses are the most common human infectious viruses and are considered essentially harmless. Compared to other human viruses, they are characterized by enormous diversity, and recombination is suggested to play an important role in their diversification and evolution. Here, by analyzing the composition of the plasma anellome of 50 blood donors, we find that recombination is also a determinant of viral evolution at the intradonor level. On a larger scale, analysis of anellovirus sequences currently available in databases shows that their diversity is close to saturation and differs among the three human anellovirus genera and that recombination is the main factor explaining this intergenus variability. Global characterization of anellovirus diversity could provide clues about possible associations between certain virus variants and pathologies, as well as facilitate the implementation of unbiased PCR-based detection protocols, which may be relevant for using anelloviruses as endogenous markers of immune status.

**KEYWORDS** anellovirus, blood anellome, recombination, metagenomics, virome

The regular presence of viruses in all ecosystems, revealed by viral metagenomics, has led to a change in our perception of viruses. Viruses are not necessarily linked to disease, since many can be harmless or even beneficial (1–3). The human virome is defined as the set of viruses found on the surface and within the body regardless of the presence of clinical symptoms of infection (4). The viral communities present differ in

Address correspondence to José M. Cuevas, cuevast@uv.es.

The authors declare no conflict of interest.

terms of abundance and composition in different anatomical body compartments (5). Among them, anelloviruses are of particular relevance, since they constitute the most prevalent human-infective viruses (6). Although some human anelloviruses have been sporadically associated with pathological conditions (7–9), they are considered essentially innocuous (6). Indeed, potentially beneficial effects have been proposed, such as the development and maturation of the immune system in infected newborns (10).

Despite the high prevalence of anelloviruses, little is known about their biology due to the lack of appropriate cell cultures and animal models. In this field, a recent study has shown that anelloviruses can be produced *in vitro*, although the system is inefficient and unable to promote their propagation (11). The International Committee on Taxonomy of Viruses (ICTV) classifies human anelloviruses into three genera: torque teno virus (TTV; alphatorquevirus), torque teno mini virus (TTMV; betatorquevirus), and torque teno midi virus (TTMDV; gammatorquevirus). TTV, TTMV, and TTMDV have been subdivided into 26, 38, and 15 species, respectively, and these numbers are expected to increase considerably as more isolates are identified (12, 13). Recent metagenomics studies have identified thousands of complete or nearly complete anellovirus genomes (14–16). These findings have confirmed the extraordinary diversity of anelloviruses, compared to other human viruses, with recombination playing a major role (16).

Besides showing that mutualistic/commensal interactions between viruses and hosts are common, the description of the viruses present in blood has direct consequences for public health (17). For example, since anellovirus load is higher in immunosuppressed patients (18, 19), this parameter could potentially be used as a health status biomarker, not only in patients with various conditions but also in people without known pathologies (20, 21). In this study, we have used a previously described protocol for viral enrichment (12) including high-speed centrifugation, random DNA amplification, and massive sequencing to characterize the anellome of 50 plasma samples from blood donors. For this purpose, the preferential amplification of single-stranded circular DNA, characteristic of anellovirus genomes, was promoted by using the multiple displacement amplification (MDA) method (22). Next, the complete or nearly complete anellovirus genomes described in our study were used to compare prevalence and demographic factors and to analyze the influence of recombination at the intradonor level. Finally, the overall diversity of the three human anellovirus genera was reevaluated using the new sequences recently deposited in the databases.

## RESULTS

**Anellovirus prevalence.** As described in Materials and Methods, 50 plasma samples from healthy donors, 25 of each sex and ranging in age from 20 to 61 years, were analyzed (Table 1). To assess viral recovery, each sample was initially spiked with 50 PFU of bacteriophage $\phi$X174. Two blank samples were used for subsequent subtraction of potentially contaminant taxa from sequencing results. To do this, the Centrifuge software was used for taxonomic classification and Recentrifuge was subsequently employed to subtract potential contaminations from real samples. $\phi$X174, which was used as an internal control, was present in only 30 of the 50 samples (60%). This was not surprising, given that a very small amount of the control virus was added to prevent its potential overrepresentation in the sequencing results (12), and suggests that our limit of detection for the presence of small DNA viruses was around 50 PFU per sample.

Overall, anellovirus reads were detected in 43 of the 50 samples, resulting in a prevalence of 86%, where the total number of reads per sample ranged from 11,563 to 6,255,446 (Table 1). The fact that 5 of the samples had no anellovirus or $\phi$X174 reads might suggest a failure in the enrichment process, although this association was not statistically significant (Fisher's exact test, $P = 0.221$). In some samples, anellovirus reads were detected, but not $\phi$X174 reads, and vice versa. In any case, it cannot be ruled out that samples lacking anellovirus sequences could contain a small amount of such viruses.

Contigs ($\geq$1.5 kb) obtained for each sample after the assembly of the reads were analyzed by BLASTn, allowing a total of 349 contigs to be assigned within the family

**TABLE 1** Summary of the data on the samples analyzed[a]

| Sample | Sex | Age (yr) | No. of $\phi$X174 RPM | No. of anellovirus RPM | No. of anellovirus contigs |
|---|---|---|---|---|---|
| SPI3 | F | 20 | 1,013 | 11,449 | 1 |
| SPI37 | F | 24 | 0 | 128,098 | 1 |
| SPI49 | F | 24 | 59,157 | 18,323 | 2 |
| SPI13 | F | 25 | 0 | 0 | 0 |
| SPI15 | F | 25 | 0 | 188,967 | 2 |
| SPI4 | F | 26 | 2,550 | 279,199 | 5 |
| SPI32 | F | 27 | 7,574 | 280,428 | 16 |
| SPI27 | F | 28 | 1,821 | 298,467 | 4 |
| SPI39 | F | 29 | 19,428 | 241,906 | 2 |
| SPI14 | F | 30 | 22,637 | 173,814 | 2 |
| SPI10 | F | 31 | 0 | 182,117 | 2 |
| SPI44 | F | 34 | 4,378 | 172,708 | 2 |
| SPI42 | F | 35 | 0 | 0 | 0 |
| SPI26 | F | 36 | 6,742 | 230,874 | 2 |
| SPI29 | F | 38 | 3,301 | 275,741 | 6 |
| SPI5 | F | 39 | 26,452 | 39,446 | 1 |
| SPI6 | F | 40 | 0 | 36,717 | 2 |
| SPI43 | F | 41 | 0 | 330,520 | 1 |
| SPI36 | F | 44 | 3,815 | 31,575 | 4 |
| SPI40 | F | 48 | 0 | 0 | 0 |
| SPI41 | F | 49 | 1,181 | 299,975 | 9 |
| SPI24 | F | 51 | 517 | 0 | 0 |
| SPI48 | F | 57 | 0 | 30,378 | 1 |
| SPI17 | F | 60 | 0 | 252,230 | 54 |
| SPI18 | F | 60 | 3,033 | 245,214 | 4 |
| SPI22 | M | 22 | 0 | 2,433 | 2 |
| SPI35 | M | 22 | 0 | 313,202 | 14 |
| SPI38 | M | 24 | 9,218 | 205,034 | 5 |
| SPI16 | M | 25 | 0 | 192,184 | 5 |
| SPI23 | M | 25 | 511 | 151,280 | 34 |
| SPI50 | M | 25 | 1,966 | 291,777 | 9 |
| SPI30 | M | 26 | 5,276 | 111,920 | 2 |
| SPI12 | M | 27 | 9,851 | 140,769 | 15 |
| SPI8 | M | 28 | 3,394 | 158,049 | 5 |
| SPI28 | M | 31 | 1,017 | 11,896 | 2 |
| SPI31 | M | 31 | 48,655 | 0 | 0 |
| SPI45 | M | 31 | 1,766 | 276,596 | 7 |
| SPI7 | M | 37 | 1,482 | 232,221 | 10 |
| SPI9 | M | 37 | 4,648 | 224,775 | 29 |
| SPI47 | M | 37 | 0 | 23,544 | 1 |
| SPI33 | M | 39 | 0 | 0 | 0 |
| SPI2 | M | 40 | 0 | 289,084 | 4 |
| SPI46 | M | 41 | 0 | 0 | 0 |
| SPI21 | M | 44 | 5,775 | 35,225 | 3 |
| SPI34 | M | 48 | 0 | 61,978 | 6 |
| SPI11 | M | 50 | 748 | 47,463 | 5 |
| SPI25 | M | 52 | 0 | 72,786 | 10 |
| SPI1 | M | 57 | 0 | 239,825 | 28 |
| SPI19 | M | 60 | 367 | 276,924 | 28 |
| SPI20 | M | 61 | 4,486 | 89,519 | 2 |

[a]For each sample, sex (F, female; M, male), age, number of reads per million (RPM) classified in Recentrifuge (45) for the virus used as an internal control (i.e., $\phi$X174) and within the *Anelloviridae* family, and the number of contigs with a size larger than 1.5 kb obtained after assembly with MetaSPAdes (46) are indicated.

*Anelloviridae* (Table 1; see also Table S1 in the supplemental material). Sixty-five of the contigs had overlapping ends and could therefore be considered complete genomes. Given the balanced sex ratio and wide age range among donors, we evaluated the effect of these demographic factors on the number of observed anelloviruses per individual. Men contained on average 9.0 ± 2.0 contigs, versus 4.9 ± 2.2 in women. To analyze these factors, we used a generalized linear model in which the response variable was anellovirus contig count (modeled using a Tweedie distribution) and the explanatory factors were age, sex, and their interaction. This showed that the number of contigs was significantly higher in men ($P = 0.016$) and increased with age ($P = 0.005$),

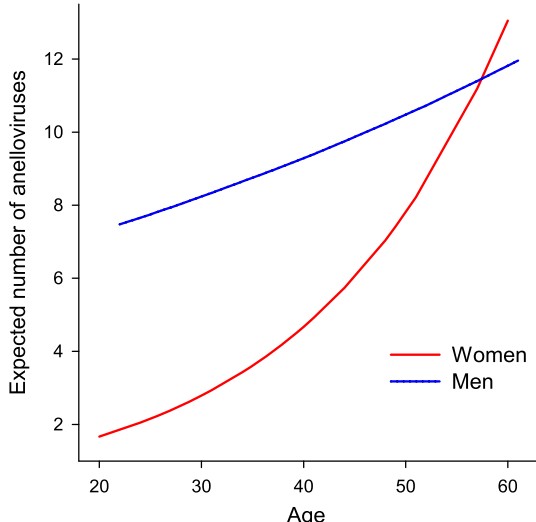

**FIG 1** Influence of demographic factors on expected anellovirus. The number of anelloviruses per individual was estimated using a generalized linear model in which the response variable was anellovirus contig count (modeled using a Tweedie distribution) and the explanatory factors were age, sex, and their interaction.

whereas the interaction term was marginally significant. The expected number of anelloviruses was approximately three times higher in young men than in young women, whereas the two sexes converged at older ages (Fig. 1). When this analysis was redone independently for each genus, the same trend was observed, but significant differences were observed only for the age factor in TTV and TTMV ($P < 0.05$) and for the sex factor in TTMDV ($P < 0.001$). We then redid the analyses using the number of anellovirus reads per million as the response variable, which could be considered a proxy for viral load (Table 1). However, no significant differences were observed either in the overall analysis or independently for each genus. This suggests that there are differences in anellovirus diversity according to age and sex but that these factors have no obvious association with viral load. Apart from anelloviruses, the taxonomic classification of reads allowed the detection of 4,213 bacteriophage reads subsequently assembled into a contig with overlapping ends of 5,930 bases, which could be considered a complete genome. This genome was classified by BLASTn into the family *Microviridae*. A BLASTp analysis of the five largest putative open reading frames (ORFs) showed homologies ranging from 87.9 to 99.7% with the closest described sequence (Table S2).

**Anellovirus diversity.** For each anellovirus contig, the ORF1 nucleotide sequence was obtained for subsequent analyses. The full-length ORF1 was obtained for all but 21 of the 349 contigs (94%). Partial ORF1 sequences were not discarded, as they included a coding region of at least 500 amino acids. A phylogenetic analysis was then carried out including the previously proposed reference species (15) and the newly described ORF1 sequences. This allowed assignment of contigs as belonging to TTV, TTMV, or TTMDV genera (197, 88, and 64 sequences, respectively) (Fig. 2, Table S1, and Fig. S1). Regarding the contigs that could be considered complete genomes, most of them (57 out of 65) belonged to the genus TTMV and only a few to the genera TTV and TTMDV (3 and 5 contigs, respectively). This may be explained by the presence of shorter GC-rich regions in TTMV (23), which may facilitate the assembly of complete genomes.

A more detailed analysis of the prevalence of the different genera in the anellovirus-positive samples showed that TTV, TTMV, and TTMDV were present in 88.4% (38 out of 43), 65.1% (28 out of 43), and 44.2% (19 out of 43), respectively (Fig. 2). Concerning the presence of coinfections, 27.9% (12 out of 43) of the samples had triple coinfections,

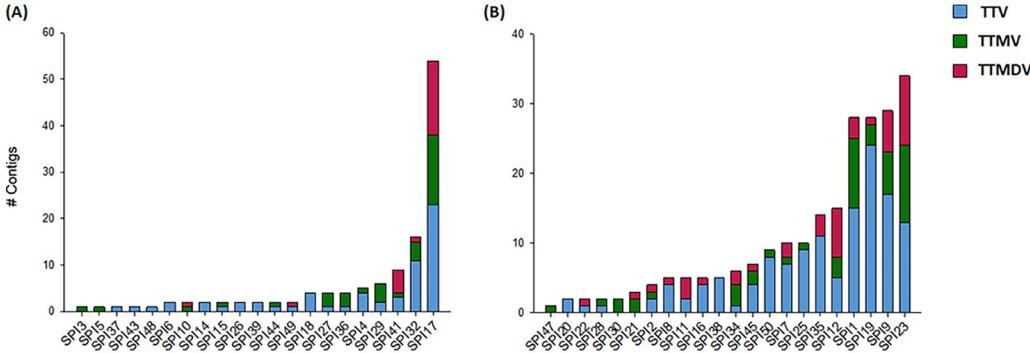

**FIG 2** Distribution of anellovirus contigs generated for female (A) and male (B) samples. For each sample, the proportion of sequences belonging to the TTV, TTMV, and TTMDV genera is shown in blue, green, and red, respectively.

41.9% (18 out of 43) had double coinfections (i.e., TTV plus TTMV, TTV plus TTMDV, or TTMV plus TTMDV), and 30.2% (13 out of 43) had monoinfections. In the last case, 10 samples showed sequences belonging exclusively to TTV, 3 samples showed sequences belonging exclusively to TTMV, and no samples showed sequences belonging exclusively to TTMDV. Overall, 69.8% of the samples showed coinfections, although this percentage increased to 86.0% if we extended the definition of coinfection to the potential presence of different intragenus variants (Fig. 2 and Table S1). Notably, more than 25 viral variants were detected in five of the samples, with triple coinfections occurring in all cases (Fig. 2).

**Intradonor recombination analysis.** It has recently been shown that the enormous variability of anelloviruses can be at least partially explained by pervasive recombination (16). To try to shed some light on the importance of this mechanism on a smaller scale, recombination analyses were carried out at the intradonor level on our data set. The requirement for a sample to be included in the analysis was that it had at least 10 viral variants for a given genus, which was met by 7 samples for TTV, 3 for TTMV, and 2 for TTMDV (Table 2). In each of these 12 cases, sequences were aligned and initially analyzed using RDP4 (24). Globally, 68 putative recombination events were detected (Table 2 and Table S3). Although the assembly of massive sequencing reads could produce artifacts, potentially jeopardizing the accuracy of recombination analysis, a previous study combining massive and Sanger sequencing supported the reliability of our pipeline (12).

After carrying out the preliminary recombination analysis using RDP4, additional analyses were performed to contrast the confidence of each recombination event independently. First, we assessed the level of phylogenetic signal in each putative recombination fragment by likelihood mapping (25). Fragments with more than 50% unresolved topologies were discarded, retaining 41 out of 68 recombination events (Table S3). Then, the tree topologies derived from the 41 recombination events were compared with the tree topology of the complete ORF1, which served as a reference. We considered a fragment recombinant if the resulting tree had a significantly higher likelihood than the reference tree using expected likelihood weights (ELW) and approximately unbiased (AU) incongruence tests (Table S4).

**TABLE 2** Summary of samples selected for recombination analysis and RDP4 output[a]

| Sample | No. of contigs for virus: | | | |
|---|---|---|---|---|
| | Anellovirus | TTV | TTMV | TTMDV |
| SPI1 | 28 | 14 (4) | 11 (3) | 3 |
| SPI9 | 29 | 17 (8) | 6 | 6 |
| SPI17 | 54 | 24 (5) | 15 (4) | 15 (7) |
| SPI19 | 28 | 24 (9) | 3 | 1 |
| SPI23 | 34 | 13 (7) | 11 (4) | 10 (8) |
| SPI32 | 16 | 11 (4) | 4 | 1 |
| SPI35 | 14 | 11 (5) | 0 | 3 |

[a]Only those cases where at least 10 sequences were present for a given genus were used in the recombination analyses. The number of putative recombination events detected by RDP4 is indicated in parentheses.

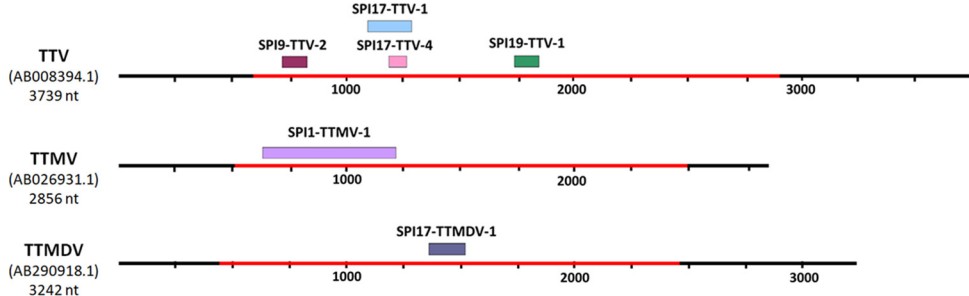

**FIG 3** Schematic representation of the recombination events detected. The six recombination events finally detected are plotted on the reference genome of each genus. The red line indicates the location of ORF1, which is the region used in recombination analyses, for each reference genome.

Only 6 recombination events out of 41 analyzed (Fig. 3 and Table S4) simultaneously met the criteria described for trees obtained from the recombinant fragment (i.e., ELW >0.5 and AU >0.05) and the ORF1 (i.e., ELW <0.5 and AU <0.05). Of these 6 events, 4 were detected in TTV and one each in TTMV and TTMDV, corroborating that recombination is common in all three anellovirus genera (16). In addition, in sample SPI17, where the highest number of anellovirus sequences was identified, 3 recombination events were detected, 2 of them in TTV and 1 in TTMDV. It is also noteworthy that, despite the small number of sequences included in each analysis, major and minor parents were identified in five of the six recombination events (Table S3). Although it cannot be ruled out that recombination occurred before infection of the analyzed individuals, our results may suggest that recombination could also have occurred during viral replication in the donor himself or herself.

The size of the 6 recombinant fragments detected was variable, ranging from 96 to 542 nucleotides (nt) (Fig. 3). A certain bias in the distribution of these recombination events was observed, as they were predominantly located in the first half of ORF1 (i.e., the N-terminal region of the protein). The ORF1 presents at the N terminus several arginine residues and some lysines, similar to the ARM motif found in some viral families, for which nuclear localization signaling and viral genome binding have been described (26). These ARM-containing viruses also often present a jelly roll domain, typical of viral capsid proteins, and the presence of this domain in the N-terminal region of ORF1 has been suggested (27). In addition, the spike domain is partially present in the first half of ORF1, which includes a hypervariable region comprising most of the spike P2 domain (27). The recombinant fragments detected included, in whole or in part, some of the above-mentioned regions. However, there is no experimental evidence to support these putative functions in anelloviruses or to explain the role of the hypervariable region.

**Global diversity analysis of human anellovirus.** Anelloviruses are an ancient family characterized by a vast diversity (16), and recently, thousands of human anellovirus sequences have been deposited in databases (14–16). It is therefore appropriate to quantify the current diversity of human anelloviruses. To this end, as described in Materials and Methods, available anellovirus sequences were downloaded, and those described in the present work were also included. ORF1 was then selected and classified as belonging to TTV ($n = 3,497$), TTMV ($n = 6,704$), and TTMDV ($n = 5,745$). For each virus genus, a matrix of pairwise sequence identities was calculated. These similarity matrices were transformed into distance matrices and used to classify sequences by hierarchical clustering with a complete linkage method. Then, a minimum of 69% of pairwise sequence identity was used as a species demarcation criterion (13) to define species clusters, and species richness and Shannon diversity for each anellovirus genus were analyzed.

The species richness observed for each anellovirus genus was close to the saturation predicted by extrapolation (Fig. 4A and Table S5). However, species richness for TTV was significantly lower than that for TTMDV and, especially, TTMV. Because diversity also depends on the relative frequency of each species in a data set, equivalent to assemblage in ecological terms, we also analyzed Shannon diversity, i.e., the effective number

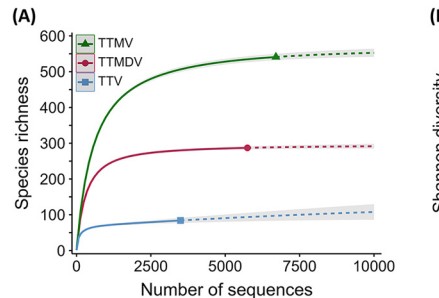
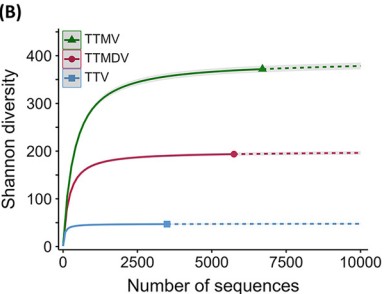

**FIG 4** Sample-size-based interpolation (rarefaction) and extrapolation curves for species richness (A) and Shannon diversity (B) of each anellovirus genus. Solid lines represent interpolation (rarefaction), dashed lines represent extrapolation, solid shapes indicate observed values, and shaded areas show 95% confidence intervals. Species richness and diversity analyses were performed with the iNEXT package in R (58).

of species based on the Shannon index. Shannon diversity estimates the number of species that would show the same Shannon index calculated for a given assemblage if they were equally common (Fig. 4B and Table S5). Similarly, Shannon diversity showed the same pattern as species richness, with TTV markedly less diverse than the other two genera. It should be noted that the difference in the behavior of the predicted sample-size-based curves for species richness and Shannon diversity of TTV is due to the presence of 16 singletons (i.e., species defined by a single sequence) contained among the 84 species observed in the 3,497 sequences analyzed. These surprisingly abundant singletons are major contributors to species richness predictions but hardly contribute to Shannon diversity estimates. However, these sequences should be cautiously considered as they may originate from sequencing or classification artifacts or represent extreme hypermutation or heterologous recombination events with unknown fitness effects.

To assess the evolutionary causes of these differences in diversity among genera, the effect of recombination relative to mutation (r/m) was quantified. First, ORF1 sequences were clustered to remove redundancy (identity threshold of 0.95), yielding 970, 2,797, and 2,600 clusters for TTV, TTMV, and TTMDV, respectively. A total of 100 representative sequences were randomly selected 10 times, and their r/m was estimated. Recombination was the greatest contributor to diversity, with r/m estimates always greater than 600 (Table 3). Globally, statistically significant differences among the r/m rates were found, with TTMV and TTV showing the highest and the lowest rate, respectively (Table 3). In more detail, pairwise comparisons showed significant differences between TTV and TTMV and between TTV and TTMDV (Welch's $t$ test, $P < 0.001$ for both comparisons) but not between TTMV and TTMDV (Welch's $t$ test, $P = 0.120$). As expected, TTV, which is the genus showing the lowest variability in the species richness, also presented the lowest r/m rate. This result could be explained as a consequence of a lower recombination rate, a higher mutation rate, or a combination of the two.

## DISCUSSION

Anelloviruses are present with a high prevalence in human populations (28), although this varies widely in different geographical locations. For example, prevalence studies in Romanian and Qatari populations have shown values of 58.4% and 85%, respectively (29, 30). Here, we found a prevalence of 86%. These considerable differences between studies are probably a consequence of the lack of a standard anellovirus detection method. These methods are usually based on PCR amplification and are

**TABLE 3** Analysis of variance of the results obtained in the r/m analysis for the human anellovirus genera

| Genus | Mean ± SD | $F$ | $P$ |
|---|---|---|---|
| TTV | 658.467 ± 32.310 | 33.301 | <0.001 |
| TTMV | 1,059.550 ± 83.167 | | |
| TTMDV | 981.809 ± 181.102 | | |

therefore limited by the enormous variability of these viruses (10). This problem did not exist in our work, since random amplification and massive sequencing were carried out, in addition to a first step of enrichment of the viral fraction. In line with this, recent evidence suggests that anellovirus detection by metagenomics is more reliable than that by quantitative PCR-based methods (31). In addition, most of the anellovirus-positive samples in our study had coinfections with variants of two or even all three human anellovirus genera, resulting in a distinctive anellovirus profile for each subject (16). However, despite the high prevalence of anelloviruses in our study, an even higher prevalence masked by technical failures in the enrichment process cannot be ruled out.

In this study, anellome analysis was performed on individual samples, which allowed us to analyze the possible influence of sex and age on the anellome composition. On the one hand, a significantly higher number of anellovirus contigs was recovered in males. This result corroborates previous work where several explanations have been proposed, such as the tendency of women to have higher immune responses (32) or the differential effects that sex steroid hormones may have on the host immune system (33). We also found a significant effect of age, consistent with previous work (34). This increase in prevalence may be due to a decrease in the immune response due to immunosenescence (33). We noticed that sex-associated differences in the number of anellovirus contigs were more evident in younger individuals, whereas older women and men showed no differences.

The enormous diversity that characterizes anelloviruses is probably a consequence of millions of years of host-virus coevolution (10). However, rapid evolution may also play a role in their high variability, which could be modulated by two nonexclusive factors: mutation and recombination. Regarding the first factor, the replication mechanism of anelloviruses is poorly understood, as neither a culture system nor an animal model for virus multiplication is available. Since anelloviruses use a host polymerase (35), they might not exhibit high mutation rates (10), although some indirect evidence casts doubt on this assumption. On one hand, anellovirus genomes with hypermutation patterns consistent with antiviral host innate immunity exerted by APOBEC3 proteins have been described (36). However, it has been suggested that this antiviral defense mechanism does not play a major role in viral evolution but contributes to the stability of the anellome (36). On the other hand, the ORF1 of anelloviruses has conserved motifs similar to Rep protein motifs, present in the *Circoviridae* and *Geminiviridae* families. In these families, Rep proteins interact with cellular DNA polymerases, affecting their copy fidelity (37). Regarding recombination, it requires that different virus variants coinfect the same cell to create new genetic variation, but this seems a relatively plausible scenario in light of the results obtained by us and others. Accordingly, clear evidence of recombination has been described in human (16) and nonhuman (38) anelloviruses. In our study, the analysis of relatively small sets of sequences, corresponding to intraindividual variability, has also shown the presence of significant recombination events, which corroborates the influence of this mechanism in the global variability of anelloviruses (10). These recombination events were located in the first half of the ORF1, which is consistent with previous results showing intragenomic rearrangements in TTV (39). In addition, evidence of recombination in noncoding regions has also been shown, suggesting that this phenomenon may occur throughout the genome and not only in regions involved in immune evasion (16, 40). Furthermore, our global analysis of the thousands of ORF1 sequences currently available revealed clear differences in species richness and diversity among the three anellovirus genera (Fig. 4). These differences were congruent with variations in the relative contribution of recombination to mutation.

The three genera of human anelloviruses share clear homology at the genomic level but also differ in length. However, until recently, little attention has been paid to these differential characteristics. Recent work has now delved into the enormous diversity of these viruses and has shown clear differences between the three genera (14, 16). In this new context, our work suggests that these differences are primarily a consequence of recombination. However, despite recent efforts (11), there is still an urgent

need to implement experimental systems capable of propagating these viruses in cell culture and/or animal models and to analyze in the laboratory the molecular mechanisms that govern their evolution. A clear example of the limited information available for this family of viruses is that even today, it is still postulated that TTV may be a naked DNA particle without a virion structure (41). The development of new experimental tools will shed light on the biological mechanisms that explain the differences in diversity between the three genera, which could be conditioned, at least in part, by potential differences in mutation and/or recombination rates. Moreover, since anellovirus diversity is largely modulated by the effect of recombination, the use of strictly clonal models of evolution may not be adequate to infer relationships or distances between sequences (16). Consequently, the taxonomic classification of anelloviruses is an unprecedented challenge that requires the implementation of new analytical methodologies that take into account their enormous diversity and reticulate evolution.

## MATERIALS AND METHODS

**Sample collection.** A total of 50 plasma samples from healthy donors (SPI1 to SPI50) were collected from the Centro de Transfusiones de la Comunidad Valenciana (Valencia, Spain) from 28 September to 19 October 2020 and stored at $-80°C$ until use. Following the Declaration of Helsinki, all subjects provided written informed consent. The protocol was approved by the University of Valencia ethics committee (IRB no. H1489496487993). Each sample analyzed in this study consisted of 10 mL of plasma from a single donor. Half of the samples were obtained from women with an age range between 20 and 60 years and the other half from men with an age range between 22 and 61 years (Table 1).

**Viral enrichment, DNA extraction, and random amplification.** To assess viral recovery, each plasma sample (10 mL each) was spiked with 50 PFU of bacteriophage $\phi$X174, a circular single-stranded virus. The concentration and purification protocol has been described previously (12), with the exception that nucleic acid extraction here focused on the DNA fraction, as it was intended to favor anellovirus detection. Briefly, plasma samples were filtered through 1.0-$\mu$m filters to remove cells and other nonviral particles and subjected to high-speed centrifugation (87,000 $\times$ $g$, 2 h, 4°C), washed with 1$\times$ phosphate-buffered saline (PBS) (87,000 $\times$ $g$, 1 h, 4°C), and resuspended in 245 $\mu$L 1$\times$ digestion buffer (Turbo DNA Free kit; Ambion). Then, 5 $\mu$L of Turbo DNase, 2 $\mu$L of Benzonase (Sigma), and 2 $\mu$L of micrococcal nuclease (New England Biolabs [NEB]) were added to the sample to remove the remaining unprotected nucleic acids. After incubation (1 h, 37°C), 20 $\mu$L of stop reagent was added, following the manufacturer's instructions. Then, 240 $\mu$L supernatant was transferred to a new tube and the entire volume was used for DNA extraction with the QIAamp viral RNA minikit. Random amplification was carried out with the TruePrime whole-genome amplification (WGA) kit (Sygnis). To identify possible environmental contaminants in the materials and reagents, two blank samples containing 10 mL 1$\times$ PBS were processed in parallel with the rest of the samples.

**Massive parallel sequencing.** Each product from the random amplification step was used for library preparation using the Nextera XT DNA library preparation kit with 15 amplification cycles (Illumina) and subjected to paired-end sequencing in a NextSeq device with the read length of 150 bp at each end.

**Sequence analysis.** Sequence data were quality checked using FastQC v0.11.9 (http://www.bioinformatics.babraham.ac.uk/projects/fastqc/, accessed on 22 November 2022) and MultiQC v1.8 (42). Reads were deduplicated using clumpify.sh and quality filtered using bbduk.sh, both tools from BBTools suite v38.82 (43). A quality trimming threshold of 20 was used, and reads below 70 nucleotides in length were removed from the data set. Sequence identification was performed with the Centrifuge software package version 1.0.4 (44) using a minimum exact match of 18. A customized database was generated from the NCBI nucleotide database downloaded in September 2020. The Centrifuge download tool was used for incorporating archaeal, virus, bacterial, and fungal genomes from the September 2020 RefSeq database at the "Complete Genome" and "Chromosome" assembly levels. Centrifuge results were postprocessed for contaminant removal, i.e., subtraction of taxa present in blank controls, with Recentrifuge version 1.3.2 (45) using a minscore of 22. Regardless of the taxonomical classification of the individual reads, assembly was individually performed for each sample with metaSPAdes version 3.15.0 (46) using default parameters. This is effective for the detection of new anelloviruses as it avoids unintentional loss of viral reads (47). Sequence identity analysis of the contigs was performed against a local copy of the NCBI nucleotide database using BLASTn v2.10.0 with an E value cutoff of $10^{-5}$.

Putative open reading frames were identified using ORF Finder (https://www.ncbi.nlm.nih.gov/orffinder/, accessed on 22 November 2022).

**Influence of sex and age on anellovirus contig counts.** A general linear model was used to test the effects of gender and age on the observed contig count. A Poisson model was discarded because the dependent variable was highly overdispersed (variance $>>$ mean). Since the observed variable contained a large fraction of zeros but also some large values, we used a Tweedie distribution model. We also explored a negative binomial distribution, which led to similar conclusions (significant effects of age and sex). The canonical log link function was used. These statistical analyses were performed in SPSS version 26 software.

**Recombination analysis.** Samples with at least 10 viral variants for a given genus were used in the recombination analyses, which were performed at the intradonor level. The nucleotide sequences of

ORF1 were aligned with MAFFT v.7 (48) (E-INS-i option) for each genus separately. This was done to avoid artifacts due to high sequence divergence between the three anellovirus genera. Then, alignments were analyzed with RDP4 (24) using seven detection methods (RDP, GENECONV, BootScan, MaxChi, Chimaera, SiScan, and 3Seq) to identify putative recombination events. Only those putative recombination events detected by at least three out of the seven methods implemented in RDP4 were accepted. Putative recombinant fragments were extracted from the alignment using the extractalign tool (https://www.bioinformatics.nl/cgi-bin/emboss/extractalign, accessed on 22 November 2022). The level of phylogenetic signal of each partial alignment was assessed using likelihood mapping with the evaluation of 1,000 random quartets using IQ-TREE v1.6.12 (49). Fragments with less than 50% of unresolved quartets were kept and used to test the phylogenetic incongruence of the recombinant fragment (50). To do so, IQ-TREE v.1.6.12 was used to build maximum likelihood trees of each partial alignment and from the complete ORF1 under the best evolutionary model (51). Both phylogenies were assessed for phylogenetic incongruence using the expected likelihood weights (ELW) (52) and the approximately unbiased (AU) (53) tests. On one hand, the ELW test assigns a probability to each tree (up to 1) based on its likelihood given the alignment of the recombinant fragment. On the other hand, the AU test provides the $P$ value of each tree, so if this is less than 0.05, the tree does not present sufficient statistical evidence to explain the data, and therefore, it is rejected. Given this, we considered potential recombinants those events in which the ELW value was greater than 0.5 for the recombinant tree (54) and the AU test rejected the topology of the complete ORF1 (reference topology) compared with the one obtained from the recombinant fragment (55).

**Analysis of the global variability of anelloviruses.** Publicly available human anellovirus sequences were retrieved from NCBI's GenBank repository (April 2022) with the following query terms: "Anelloviridae" [Organism] AND "*Homo sapiens*" [Organism]. The downloaded fasta file included 17,127 coding sequences. This file was subsequently filtered with a homemade script to select coding sequences larger than 1,500 nt. After adding the 349 sequences described in this study to the filtered file, the final file contained 15,953 sequences. Sequences were then assigned to the TTV, TTMV, and TTMDV genera using a local BLASTn search with 75% identity as a threshold against a database containing the reference sequences of each genus described by ICTV (13). For those sequences that could not be classified by this approach, a phylogenetic analysis was used. For this, a file was created that included the unclassified sequences and the reference sequences for each genus. An alignment was then performed with MAFFT v.7, and a phylogenetic tree was constructed with IQ-TREE v1.6.12. The obtained tree was annotated with EvolView (56) and used for the classification of the remaining sequences.

Pairwise identity matrices were obtained with SDT v1.2 (57) using the entire ORF1 nucleotide sequence independently for each genus, and sequences sharing more than 69% pairwise sequence identity (13) were classified as the same species by hierarchical clustering using the R *hclust* function. For that, similarity matrices were previously transformed into distance matrices (distance = 1 − sequence identity), and complete linkage was used as a clustering method to ensure that the chosen species demarcation criterion was satisfied for any pair of sequences grouped. To assess the species richness and diversity of each anellovirus genus, the number of sequences clustered within each defined species was used to create a sequence-based abundance data set that was subsequently analyzed in R using the iNEXT package (58). This package was used to perform asymptotic analyses and compute sample-size-based rarefaction and extrapolation curves of two common Hill numbers (species richness and Shannon diversity). Hill numbers (or the effective number of species) represent a well-established mathematical framework in ecology to compare species diversity across different assemblages (58, 59).

To determine the relative contribution of recombination over mutation (r/m) in the global data sets, representative sequences for each genus were first obtained using mmseqs (60) (–cov-mode 1 –id 0.95). This step was taken to prevent differential species abundance from affecting the estimates of r/m (61). Then, a total of 100 sequences were randomly selected and aligned with MAFFT v.7 (48). Alignments were used to build phylogenetic trees with IQ-TREE v.1.6.12 (49). The resulting alignments and trees were used as input for ClonalFrameML (62), and r/m was calculated using the following formula: r/m = R/theta × delta × nu, where R/theta is the ratio of recombination to mutation rate, delta is mean import length, and nu is the nucleotide distance between imported sequences (61). This process was repeated 10 times to avoid biases due to the selection of the 100 random sequences.

**Data availability.** The raw sequence reads were deposited in the Sequence Read Archive of GenBank under accession number PRJNA763062. The newly described sequences belonging to anelloviruses were deposited in GenBank under accession numbers MZ824758 to MZ825040 and MZ825088 to MZ825153 (see Table S1 in the supplemental material), and a single microvirus sequence was deposited under accession number MZ821031.

## SUPPLEMENTAL MATERIAL

Supplemental material is available online only.
**SUPPLEMENTAL FILE 1**, XLSX file, 0.03 MB.
**SUPPLEMENTAL FILE 2**, XLSX file, 0.01 MB.
**SUPPLEMENTAL FILE 3**, XLSX file, 0.03 MB.
**SUPPLEMENTAL FILE 4**, XLSX file, 0.01 MB.
**SUPPLEMENTAL FILE 5**, XLSX file, 0.02 MB.
**SUPPLEMENTAL FILE 6**, PDF file, 0.2 MB.

## ACKNOWLEDGMENTS

This research was funded by the Spanish Ministerio de Economía, Industria y Competitividad (MINECO) cofinanced by FEDER funds, grant numbers SAF2017-82287-R and PID2020-118602RB-I00, and the Generalitat Valenciana, grant number AICO/2021/085.

We declare no competing interests.

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
