## [Reviewer comments · Microbiology Spectrum]

Microbiology Spectrum

Human anelloviruses: influence of demographic factors, recombination and worldwide diversity

María Cebriá-Mendoza, Beatriz Beamud, Iván Andreu-Moreno, Cristina Arbona, Luís Larrea, Wladimiro Diaz, Rafael Sanjuán, and Jose Cuevas

Corresponding Author(s): Jose Cuevas, Institute for Integrative Systems Biology, I2SysBio, Universitat de València-CSIC

Review Timeline:

Submission Date:	January 18, 2023
Editorial Decision:	March 28, 2023
Revision Received:	April 24, 2023
Accepted:	May 5, 2023

Editor: Igor Rouzine

Reviewer(s): Disclosure of reviewer identity is with reference to reviewer comments included in decision letter(s). The following individuals involved in review of your submission have agreed to reveal their identity: Jiandong Bao (Reviewer #2)

Transaction Report:

DOI: <https://doi.org/10.1128/spectrum.04928-22>

March 28, 2023

Dr. Jose M Cuevas
Institute for Integrative Systems Biology, I2SysBio, Universitat de València-CSIC
Carrer del Catedràtic Agustín Escardino Benlloch, 9
Paterna, Valencia 46980
Spain

Re: Spectrum04928-22 (Human anelloviruses: influence of demographic factors, recombination and worldwide diversity)

Dear Dr. Jose M Cuevas:

In the revised version, kindly address the comments of the Reviewers. While you are not required to increase the sample size, as suggested by reviewer 2, you are advised to address this comment by providing a simple statistical estimate.

Link Not Available

Sincerely,

Igor Rouzine

Journals Department
Reviewer comments:

Reviewer #1 (Comments for the Author):

With great interest, I read the manuscript "Human anelloviruses: influence of demographic factors, recombination and worldwide diversity" by Cebriá-Mendoza et al. The study gives an insight into anellovirus diversity, with a main focus on recombination. Recombination in anelloviruses is an important subject to explore, and so far there are just a few studies that discuss it, especially for human anelloviruses.

The manuscript is most clearly and well-written and the study is well-designed. I have only a few remarks. The main one is regarding the discussion on recombination: it would be good to compare this study with the previous ones that showed the

recombination patterns in anelloviruses, especially with the paper by Arze et al 2021. Did they observe similar patterns - were they in the same locations of the genome and similar length, were they more/less common? And it would be also interesting to discuss why the recombination events were located mainly within the ORF1.

I would also advise the authors to read the manuscript carefully and correct the scientific writing: it contains a few typos (for instance Page 8, line 29, it should be "us" instead of "we").

List of remarks:

Page 3, line 22: please cite here also the paper by Varsani et al (reference 24 in the current reference list).

Page 4, line 29: Perhaps it would be interesting to see the number of mapped reads to the anellovirus contigs (RPM)? In order to see whether the anellovirus load (measured in RPM to make it possible to compare) is also higher in men compared to women, or is it only the difference in the number of contigs. Also interesting to see the same for the different age groups.

Page 5, line 5: Follow up to the previous remark: did the distribution of TTV vs TTMV and TTMDV change with age or gender?

Page 6, line 13: Do the authors mean the N-terminal of the ORF1? Besides, it would be interesting to roughly estimate in which domain of the ORF1 protein the fragments were located. The authors could take a look at the recent study of Liou et al [bioRxiv2022:2022.07.01.498313](https://doi.org/10.1101/2022.07.01.498313)

Page 6, line 25: I advise starting this paragraph with a short explanation of why this analysis was performed. It would improve greatly the reading experience and possibly link this paragraph with the previous one.

Page 6, line 42: The sentence: "This represents the number of species..." is rather unclear. It is not immediately clear what "This" refers to and the message is hard to decipher. Please re-write the sentence.

Page 7 line 2: "TTV markedly less diverse than other to genera". It is an interesting remark, perhaps the differences in diversity levels between anelloviruses is worth mentioning and elaborating in the discussion. It is just slightly touched in the discussion on page 8 line 36-40. Why are there such differences? Is the interpersonal diversity of TTV also smaller than TTMV/TTMDV, like it is seen for the global diversity?

Page 8 lines 36-38. "Furthermore, our global analysis of the thousands of ORF1 sequences currently available revealed clear differences in the number of species among the three anellovirus genera." I am not sure what the authors mean here: is it the level of diversity corrected for the number of species within each genus? Please clarify.

Figure 3. Perhaps add more tick marks (so more breaks in the scale) so it is easier to see where exactly the recombination events occurred (especially for TTV). Also showing where ORF1 gene is located would enhance the figure (and maybe also other ORFs, but ORF1 is the most important, since according to the authors the recombination events mainly happen within this gene)

Reviewer #2 (Comments for the Author):

This study investigated the anelloviralome of 50 healthy blood donors (25 women from 20-60 years old and 25 men range from 22-61 years old) using viral enrichment, random amplification and Illumina sequencing method, revealed 349 complete or near-complete Anelloviruse assemblies (197 TTV, 88 TTMV and 64b TTMDV) in 43 plasma samples, found 6 intra-genus recombination events in ORF1, and claimed that: 1. anelloviruses increased with age, and was approximately twice as high in men (9.0 {plus minus} 2.0 contigs) as in women (4.9 {plus minus} 2.2). 2. Recombination was the main factor promoting diversity between genera integrating thousands of public available anellovirus sequences.

Main comments

1. The small size (50 healthy blood donors) and quality of samples (only 60% internal control virus was detected, and total reads per sample vary ranged from 11,563 to 6,255,446) are insufficient to support the conclusion like "The number of detected anelloviruses increased with age, and was approximately twice as high in men as in women". Suggest increasing the sample size or adding evidence from published dataset.

2. The publicly available human anellovirus sequences were collected until April 2022, which is almost one year ago. I noticed that several new studies regarding human anellovirus have been published in 2022 and 2023 (like PMID: 35575554, PMID: 35575554, PMID: 36042550, PMID: 36726484 and PMID: 35758682). To support the title "Human anelloviruses: influence of demographic factors, recombination and worldwide diversity", suggest to add these new anellovirus sequences.

3. In citation, only two papers published in 2022 have been cited. The last paper regarding anellovirus should be cited.

4. Any new species was found in this study? And how about the distribution of anellovirus sequences at species level?

Staff Comments:

Preparing Revision Guidelines

Please return the manuscript within 60 days; if you cannot complete the modification within this time period, please contact me. If you do not wish to modify the manuscript and prefer to submit it to another journal, please notify me of your decision immediately so that the manuscript may be formally withdrawn from consideration by Microbiology Spectrum.

Response to Reviewer #1

With great interest, I read the manuscript "Human anelloviruses: influence of demographic factors, recombination and worldwide diversity" by Cebriá-Mendoza et al. The study gives an insight into anellovirus diversity, with a main focus on recombination. Recombination in anelloviruses is an important subject to explore, and so far there are just a few studies that discuss it, especially for human anelloviruses.

The manuscript is most clearly and well-written and the study is well-designed. I have only a few remarks. The main one is regarding the discussion on recombination: it would be good to compare this study with the previous ones that showed the recombination patterns in anelloviruses, especially with the paper by Arze et al 2021. Did they observe similar patterns - were they in the same locations of the genome and similar length, were they more/less common? And it would be also interesting to discuss why the recombination events were located mainly within the ORF1.

Perhaps it was not reflected in the text that the recombination analyses were done exclusively using the ORF1 alignments. Using ORF1 is a common procedure, both for classifying anellovirus species and for performing recombination analyses (for example, Arze's work, mentioned by the reviewer, also looked primarily at ORF1, in addition to a specific analysis of the 5'UTR), as the number of complete genomes in the databases is much smaller than the number of ORF1s, which covers about 60-70% of the genome of each of the three genera of human anelloviruses. We have therefore tried to clarify this in the text to avoid the confusion.

Furthermore, the methodology employed in Arze's work to look for evidence of recombination (i.e. excess repeat mutations, shifting topologies in constructed phylogenetic trees, and low linkage disequilibrium measurements) is different from ours. These methods provide global evidence of recombination but do not indicate specific recombination points, such as those shown in our work. Therefore, we cannot make the comparison proposed by the reviewer. Instead, we have now mentioned in the current version the few papers on anelloviruses that have shown evidence of recombination.

I would also advise the authors to read the manuscript carefully and correct the scientific writing: it contains a few typos (for instance Page 8, line 29, it should be "us" instead of "we").

The manuscript has been proofread and we hope that all typos have been corrected.

List of remarks:

Page 3, line 22: please cite here also the paper by Varsani et al (reference 24 in the current reference list).

OK. Done.

Page 4, line 29: Perhaps it would be interesting to see the number of mapped reads to the anellovirus contigs (RPM)? In order to see whether the anellovirus load (measured in RPM to make it possible to compare) is also higher in men compared to women, or is

it only the difference in the number of contigs. Also interesting to see the same for the different age groups.

We have redone demographic analyses using also anellovirus RPMs and these results are now shown in the main text. In line with the reviewer's comment, to facilitate comparison between samples, table 1 has been modified to indicate viral RPM instead of the total number of viral readings.

Page 5, line 5: Follow up to the previous remark: did the distribution of TTV vs TTMV and TTMDV change with age or gender?

Following on from the previous comment, we have also included in the current version the demographic analyses independently for each anellovirus genus, both for the number of contigs and RPMs.

Page 6, line 13: Do the authors mean the N-terminal of the ORF1? Besides, it would be interesting to roughly estimate in which domain of the ORF1 protein the fragments were located. The authors could take a look at the recent study of Liou et al bioRxiv2022:2022.07.01.498313

This sentence has been clarified in the text and, in addition, the Discussion section now includes additional comments on the association between the ORF1 protein domains and the recombinant fragments we have found.

Page 6, line 25: I advise starting this paragraph with a short explanation of why this analysis was performed. It would improve greatly the reading experience and possibly link this paragraph with the previous one.

A short explanation has now been added before the description of this analysis.

Page 6, line 42: The sentence: "This represents the number of species..." is rather unclear. It is not immediately clear what "This" refers to and the message is hard to decipher. Please re-write the sentence.

The sentence has been rewritten to make it easier to understand.

Page 7 line 2: "TTV markedly less diverse than other to genera". It is an interesting remark, perhaps the differences in diversity levels between anelloviruses is worth mentioning and elaborating in the discussion. It is just slightly touched in the discussion on page 8 line 36-40. Why are there such differences? Is the interpersonal diversity of TTV also smaller than TTMV/TTMDV, like it is seen for the global diversity?

To date, little attention has been paid to studying the differences between the three genera of human anelloviruses, given their apparent non-pathogenicity and the absence of experimental data. To delve deeper into the evolutionary mechanisms that explain these differences, as already mentioned in the discussion (now with more emphasis), it is necessary to establish cellular systems or animal models capable of allowing their propagation.

Page 8 lines 36-38. "Furthermore, our global analysis of the thousands of ORF1 sequences currently available revealed clear differences in the number of species among the three anellovirus genera." I am not sure what the authors mean here: is it

the level of diversity corrected for the number of species within each genus? Please clarify.

Throughout the text, we have used the expression "number of species" to refer to "species richness", but in the current version, this equivalence has been omitted to avoid confusion. However, we maintain the equivalence between "the effective number of species" and "Shannon diversity", which was already explained in the text and is a commonly used terminology in ecology.

Figure 3. Perhaps add more tick marks (so more breaks in the scale) so it is easier to see where exactly the recombination events occurred (especially for TTV). Also showing where ORF1 gene is located would enhance the figure (and maybe also other ORFs, but ORF1 is the most important, since according to the authors the recombination events mainly happen within this gene)

As mentioned in the response to the first reviewer's comment, the recombination analyses were restricted to ORF1, which is now indicated in the figure caption. In addition, during the process of modifying the figure, as proposed by the reviewer, we realised that there was an error in the location of the recombination points, which has now been corrected in the new version. In any case, the figure is merely illustrative of the results, but the actual data on the location of the recombination events are given in Supplementary Table S3.

Response to Reviewer #2

This study investigated the anellome of 50 healthy blood donors (25 women from 20-60 years old and 25 men range from 22-61 years old) using viral enrichment, random amplification and Illumina sequencing method, revealed 349 complete or near-complete Anelloviruse assemblies (197 TTV, 88 TTMV and 64b TTMDV) in 43 plasma samples, found 6 intra-genus recombination events in ORF1, and claimed that: 1. anelloviruses increased with age, and was approximately twice as high in men (9.0 {plus minus} 2.0 contigs) as in women (4.9 {plus minus} 2.2). 2. Recombination was the main factor promoting diversity between genera integrating thousands of public available anellovirus sequences.

Main comments

1. The small size (50 healthy blood donors) and quality of samples (only 60% internal control virus was detected, and total reads per sample vary ranged from 11,563 to 6,255,446) are insufficient to support the conclusion like "The number of detected anelloviruses increased with age, and was approximately twice as high in men as in women". Suggest increasing the sample size or adding evidence from published dataset.

We do not understand the reviewer's concern about sample size, as this could be a problem if we did not detect statistically significant differences in the analyses of demographic variables. However, this is not the case and our results corroborate those obtained in previous studies, as cited in the text (Moustafa et al., 2017, Haloschan et al., 2014, Focosi et al., 2020).

Regarding the concern about the quality of the samples, based on the poor detection of our control virus, we would like to point out that most studies in viromics do not use

internal controls. We believe that this is a convenient procedure and it helps us to estimate the detection limit of the technique without overly compromising the "loss" of reads associated with the control virus. Nevertheless, no statistical association was observed between the absence/presence of the control virus and the absence/presence of anellovirus, as indicated in the text, which supports the reliability of our results.

2. The publicly available human anellovirus sequences were collected until April 2022, which is almost one year ago. I noticed that several new studies regarding human anellovirus have been published in 2022 and 2023 (like PMID: 35575554, PMID: 35575554, PMID: 36042550, PMID: 36726484 and PMID: 35758682). To support the title "Human anelloviruses: influence of demographic factors, recombination and worldwide diversity", suggest to add these new anellovirus sequences.

We recognise that the sequences extracted from the databases date back a relatively long time. However, this delay is due, at least in part, to a computational problem inherent in the analysis of the samples, since obtaining the identity matrices requires the use of SDT software (the currently recommended software for anellovirus classification), which unfortunately is computationally inefficient for large matrices. Therefore, adding the currently available sequences requires redoing the identity matrices, which would involve several months of computation, thus only partially addressing the concern raised by the reviewer. In addition, a new database search has shown that the new sequences available (March 2023) represent a modest 4.5% increase over those used in our work. Because of the above, we consider that the effort involved in redoing the analyses would not address the initial concern and would fundamentally delay the publication of the paper without significantly increasing the relevance of the work.

3. In citation, only two papers published in 2022 have been cited. The last paper regarding anellovirus should be cited.

We have searched for recent anellovirus articles and added several citations in the new version of the manuscript.

4. Any new species was found in this study? And how about the distribution of anellovirus sequences at species level?

Indicating the number of new species is something we have done in previous work (Cebriá et al., 2021, *Scientific Reports; Viruses*). However, as we discuss in the last paragraph of the discussion, conventional taxonomic classification tools can be seriously affected by the considerable effect of recombination. For this reason, we prefer to avoid providing concrete information on species numbers and propose the implementation of new analytical methodologies that take into account the reticulate evolution of these viruses.

May 5, 2023

Dr. Jose M Cuevas
Institute for Integrative Systems Biology, I2SysBio, Universitat de València-CSIC
Carrer del Catedràtic Agustín Escardino Benlloch, 9
Paterna, Valencia 46980
Spain

Re: Spectrum04928-22R1 (Human anelloviruses: influence of demographic factors, recombination and worldwide diversity)

Dear Dr. Jose M Cuevas:

Your manuscript has been accepted, and I am forwarding it to the ASM Journals Department for publication. You will be notified when your proofs are ready to be viewed.

Sincerely,

Igor Rouzine
Editor, Microbiology Spectrum
